# A Bibliometric Review of Education for Sustainable Development, 1992–2022

Chuang Yang [1] and Qi Xiu [2,*]

1  Faculty of Education, East China Normal University, Shanghai 200062, China; ecnu_ych@163.com
2  College of Teacher Education, South China Normal University, Guangzhou 510631, China
*  Correspondence: qi.xiu@stu.ecnu.edu.cn

**Abstract:** The United Nations promulgated Agenda 21 in 1992, thereby recognizing education as the pathway to a more sustainable future. The launch of education for sustainable development (ESD) activities and the growing number of related studies have created an urgent need for a thorough and comprehensive review of the field. Based on 2779 publications from the related literature in the SSCI index from 1992 to 2022, this study outlines the growth characteristics, research areas, and research methods, and conducts a statistical analysis of the contributing forces of countries, institutions, and authors to demonstrate that the literature is mainly generated in developed countries. Meanwhile, this study identifies ten core journals and finds that highly productive institutions are characterized by close relationships and long-term in-depth research and collaboration with authors. Finally, a combination of Latent Dirichlet Allocation (LDA) theme models, manual coding, and pyLDAvis visualization tools identified five research themes, including foundations of ESD research, environmental education, higher education for sustainable development, sustainable development capacity, and educational technology innovation. The intellectual structure of research in this emerging interdisciplinary field is revealed, and this study provides a reference for scholars in this discipline.

**Keywords:** sustainable development education; bibliometric review; sustainability

## 1. Introduction

All countries are paying increasingly close attention to sustainable development [1]. Threats to the future of the planet, such as climate change, global warming, resource depletion, desertification, water and air pollution, high carbon emissions, food shortages, and others, call for more sustainable social structures, ways of living, and economic systems [2]. In the framework of globalization and the approach of the Millennium Development Goals set by the United Nations in 2000, sustainability has been established as one of the key goals. In the year 2015, however, the Millennium Development Goals fell short of expectations [3]. Since the Sustainable Development Goal has been established, the United Nations has decided to make this one of its top priorities. This has tremendously encouraged the publication of content that is connected to sustainable development within the scope of the 2030 Agenda for Sustainable Development [4]. The new global sustainable development framework redefines the objectives of the international community to work together to assure the better future of people and the planet, and the 2030 Agenda serves as the social and economic agreement of the 21st century [5]. The 2030 Agenda includes 17 Sustainable Development Goals (out of 169 in total). These objectives span a wide spectrum of social and economic development challenges and are global, universal, interrelated, and inclusive. One of the key objectives is education, which is crucial for accomplishing Sustainable Development Goal 4 of the 2030 Agenda and is also the cornerstone of a sustainable world [6]. The main idea of SDG 4 is to provide high-quality education for all teenagers, thereby ensuring that all learners acquire the knowledge and skills they need



to promote sustainable development [7]. Although education itself cannot bring a more sustainable future, sustainable development without support from education cannot create a sustainable future [8].

In actuality, most people do not connect sustainable development with society-related fields, especially with the education-related field, which is the key motivation system of social reform. In 1987, the World Commission on Environment and Development (WCED) established the Brundtland Commission with the aim of calling for countries to raise global awareness together of issues related to sustainable development and to act swiftly to identify and implement solutions to realize sustainable development [9]. However, at this stage, people around the world have not realized the importance of education in sustainable development [10]. The turning point came during the Earth Summit in Rio de Janeiro in 1992, when the United Nations posted Agenda 21. This document recognized that education is the better way to a more sustainable future [11]. Since then, the critical role education plays in sustainable development has been recognized around the world. At the same time, this led to a great reform in understanding sustainable development, as more and more people have realized that sustainable development should consider not only environmental and economic factors but also society and education. This is the unavoidable development style [12]. Since then, activities of education for sustainable development (ESD) have been created more and more frequently. The Johannesburg World Summit declared that the decade from 2005 to 2014 is the Decade of Education for Sustainable Development (DESD) [13,14]. In this decade, UNESCO has carried out activities of education for sustainable development globally, which have made a significant contribution to promoting the influence of educational sustainable development [15]. Recently, there has been an explosion in research output since UNESCO has recommended the concept of education for sustainable development, and great progress has been made in many fields. However, there is still dissent about sustainable development [4] and related issues, such as research method design, policy construction, implementation strategy, new environment exploration, and educational technology integration for sustainable development, which still need more research [16]. It is essential to conduct a thorough and comprehensive assessment of this field. A critical review of this field's history and progress, current debates and controversies, research gaps, and future research directions all need to be conducted. Therefore, this study conducted a literature search according to the systematic evaluation Preferred Reporting Items for Systematic Reviews and Meta-Analyses (PRISMA) report standard and obtained 2779 SSCI characteristic publications from 1992 to 2022. A retrospective study on education for sustainable development was conducted based on bibliometrics.

## 2. Literature Review

There is a paucity of literature reviews on ESD. Since the year 2015, when education was formally introduced and included as an important part of the SDGs, there is still a lack of comprehensive research on ESD. Grosseck et al. [2] conducted a statistical survey of publications and found that relevant research was mainly published about educational research (60%), green sustainable science technology (31%), environmental studies (23%), and environmental sciences (17%). As of 2004, only 37 publications were found. According to this research, 1073 papers were produced during the period from 2004 to 2018, and there is an obvious increasing trend after releasing the SDG 4 in 2015. In addition, global, large-scale reviews continue to be lacking. Most of the studies have only conducted review studies in a single field [17], for a certain journal [18], or about a single topic [19], and most of these reviews have just explored the external characterization of the literature, including the number of publications, source countries or organizations, source journals, high-yield authors, etc. However, they lack in-depth analysis of citation characteristics, and they also lack content-based thematic analysis. This limitation is partly due to the broad scope and the complex and changing nature of ESD [20]. Although there is an obvious limitation in the field of ESD literature reviews, previous studies have provided

methodological references and research foundations for using bibliometrics to evaluate ESD research outcomes. The first substantive bibliometric analysis of ESD was conducted by Wright [21]. This study provided a statistical description of the ESD-related literature for the period of 1990–2005, but it only focused on the studies in the Education Resources Information Center (ERIC) database. It is claimed that in this period, the number of articles increased, but not significantly. The number of related journals also has a similar performance, but the number of authors has significantly increased. Meanwhile, it is found that ESD-related studies appeared in cross-disciplinary and interdisciplinary journals in addition to traditional education journals, attesting to the broad interests of this kind of study. In subsequent studies, the authors extensively used bibliometric methods to review ESD and conducted bibliometric studies on specific topics, such as innovation and entrepreneurship, digital technology, school environment, university governance, and career planning for ESD [22–26].

Some scholars view ESD only in terms of specific academic areas (e.g., environment or health), essentially using the term "environmental education" to refer to ESD. However, Maurer surveyed students and found that there was a significant difference in their understanding of ESD and environmental education, suggesting that "ESD and environmental education are either overlapping concepts or two very different philosophies" [27]. This overlap and the difference between EE and ESD are considered to be the focus of attention in the bibliometric analysis, and appropriate search terms need to be selected to maximize coverage of ESD research results [28]. When necessary, manual screening is needed to identify the more research-worthy literature [2]. The concept of ESD is relatively vague and broad, and consensus is still lacking [27]. As Reid [28] notes, this lack of conceptual clarity and conflation affects the results of bibliometrics and is detrimental to the future of research development. Sorting out the history of scholarship requires clarity of concepts and understanding of research content. However, no studies prior to 2007 focused on this distinction, resulting in incomplete coverage of retrospective studies of sustainable development.

At the same time, studies have mostly focused on the external characteristics of the literature (source journals, collaborating countries, authors, etc.) and less on the content of the literature. Finally, relevant studies also point to the obvious interdisciplinary integration and thematic, cross-cutting nature of ESD. For example, Cullen's 2017 study of the literature on business, management, and economics found that these disciplines show a strong interest in sustainable development, with an exponential increase in the number of relevant studies and citations [29]. Future research on ESD is likely to be found in relation to a variety of topics across disciplines, and there will be targeted journals and sections to publish ESD research [30]. The bibliometric analysis of ESD cannot be limited to the field of education or to a few designated journals. Research on ESD should cover a wider range of sources, and it requires diverse and complex research samples and designs.

Providing scientific concepts and technical tools for measuring scientific output, bibliometrics is the interdisciplinary science of quantitative analysis of all knowledge vectors using mathematical statistical methods. Bibliometrics provides support for scientific evaluation and decision making, enables management of research performance, helps measure researchers' academic performance, analyzes characteristics of academic collaboration networks, uncovers potential research themes and hot topics, and demonstrates the possibility of multidimensional academic evaluation [31]. With the advent of online bibliographic systems, more and more information about the literature can be retrieved. However, the widespread use of retrieval systems rests more on the interpretation of the external characteristics of the literature, and the evolution of research themes and research trends are not sufficiently explored [32]. The production of knowledge depends on the development of new research and its disclosure to the scientific community; by using bibliometrics, a dynamic analysis of the scientific production in a given field can reveal more information and allow a more accurate description of the development of the field [33]. Although bibliometric methods are not perfect tools in all fields or in all

cases, there is a large literature confirming the fact that bibliometric analysis is used as a tool by the scientific community to evaluate research quality [34], to evaluate journal quality [35], to map disciplinary knowledge networks, to construct maps of academic collaborations, to describe national, institutional, and scholarly research relationships [36], and to conduct meta-analysis research [37]. This kind of method is applied not only for macro-analysis, such as national-level analysis, but also for micro-observations, such as at the institutional level.

Previous studies have explored the development of ESD and conducted an overview of ESD research based on bibliometric analysis, but most of these studies focus on review studies of a single research area or research topic. Moreover, these studies do not indicate whether the relevant scientific publications are generating new knowledge, and the understanding of research themes is still lacking. Although some researchers have focused on identifying research themes, most of these studies are based on the subjective knowledge of the researchers about the themes, and the scientificity of the results is doubtful. This study will expand the scope of research and adopt more scientific text analysis methods in order to provide more valuable findings.

In this study, Web of Science is used as the data source to analyze the progress of international research on ESD with a focus on "research overview", "research forces", and "research themes". Three main research questions are suggested, as follows:

Question 1: What is the overall volume and the main research methods chosen in the ESD research from a broader historical perspective?

Question 2: What are the major research forces of the ESD research, including countries (regions), institutions, and productivity of authors and journals?

Question 3: What are the intellectual structures of the knowledge and the key research areas of the ESD research?

## 3. Methodology

### 3.1. Data Resource

WoS is the world's largest comprehensive scholarly information resource covering the largest number of disciplines; it conveys 90% of the valuable information in the literature. It is an extremely rigorous and efficient source of literature, and it is therefore widely used in bibliometric analysis [38]. Taking into account the high matching degree between the analysis objectives and the search objectives, and the requirement for data accuracy in addition to the professionalism of the literature covered by the database, research from the Web of Science (WoS) database was selected as the data resource. Three highly relevant key terms, including sustainability education, education for sustainability, and education for sustainable development, were selected to conduct the search. These three concepts are often mixed; however, they are all the key terms of related research, thereby avoiding a situation where the research in this field is not fully covered. In addition, to ensure the operability of the textual analysis, only English publications were selected, and the SSCI and SCIE indexing databases were used to ensure the integrity and uniformity of the literature data. Furthermore, these databases were used in order to expand the range and to make sure the databases covered as many important types of literature (articles, reviews, early access, editorial materials, proceedings papers) as possible, to obtain valuable information comprehensively.

### 3.2. Research Method

Primarily based on bibliometrics, this study identifies the most important publications, authors, journals, institutions, and countries with major research contributions. Furthermore, this study examines the research hotspots and frontiers of ESD based on the content of the literature. It reveals the characteristics of ESD research in an objective, scientific, quantitative, and intuitive manner, effectively avoiding the drawbacks of subjective generalization and summarization.

Both R and Python were used for data collection, data cleaning, data analysis, data visualization, etc. R and Python are freely available software, and they are maintained by users worldwide, thereby providing a large number of packages. The cleaning of the literature data was mainly performed through the dplyr package in R; the production of relevant statistical graphs relied on ggplot2; and the identification of and production of statistics regarding basic information from the literature mainly relied on the bibiometrix package [39].

In this study, the LDA (Latent Dirichlet Allocation) theme model was used to explore the topic of education for sustainable development. The English deactivation words that come with the nltk package of Python were used as the basis, and high-frequency words, search words, unintentional words, and general words (abstract, conclusion, indication, etc.) were used as deactivation words according to the research aims. Python was used to complete the word separation, and the LDA topic modeling of ESD documents was realized using the LDA Model function. Finally, the interactive visualization of topic class clustering results was drawn with the pyLDAvis package.

## 4. Data Analysis and Findings

This study was conducted on 7 March 2023 based on the WoS database, and a total of 4541 relevant data were obtained according to the search strategy. The initial screening of the 3173 documents was performed by refining the WoS database according to the type of document, language, and research area required for this study. Moreover, in order to improve the thematic relevance of the research documents, the authors double checked the titles and abstracts of the documents and the duplicates; missing fields and publications focused on "ecology", "climate", and "resources" that were not related to ESD were deleted. A total of 394 unqualified documents were removed, and a final database of 2779 documents was obtained. The data were further normalized according to the PRISMA (shown in Figure 1) standard, and duplicate publications, irrelevant publications, and publications severely missing data were removed. At last, 2779 valid data were obtained.

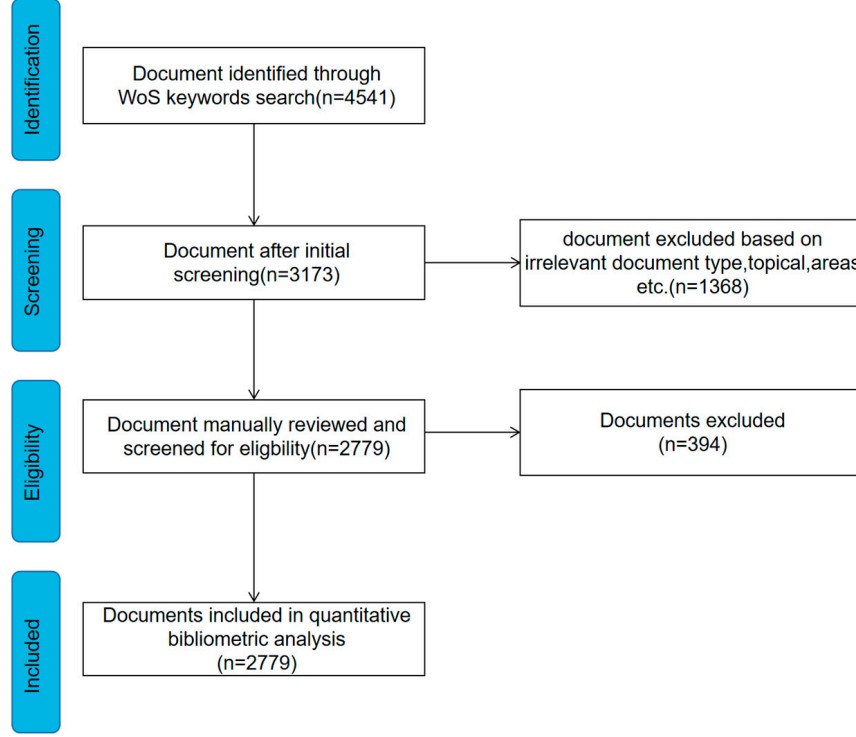

**Figure 1.** PRISMA diagram describing the collection of documents from WoS.

### 4.1. Overview of the Related Studies

The compound annual growth rate (CAGR = (Ending Value/Beginning Value) ^(1/years) − 1) of the literature from 1992 to 2022 was 23.11%. This study set the starting time as 1992 according to the development history of ESD, but the search results showed that the first related study recorded in the WoS database was published in 1993. This may be related to the fact that the concepts of sustainable development and education for sustainable development had just emerged. In addition, the concept of sustainable development was only formally introduced by the United Nations in 1996, and a lack of consensus may have been a problem for terminology expressions for researchers before then. For the whole development process, this problem does not affect the analysis results; as shown in Figure 2, the relevant literature was very scarce in the early period, and the topic did not attract academic attention until the year of 2005. This is in line with the findings of Gabriela Grosseck (2019) [2].

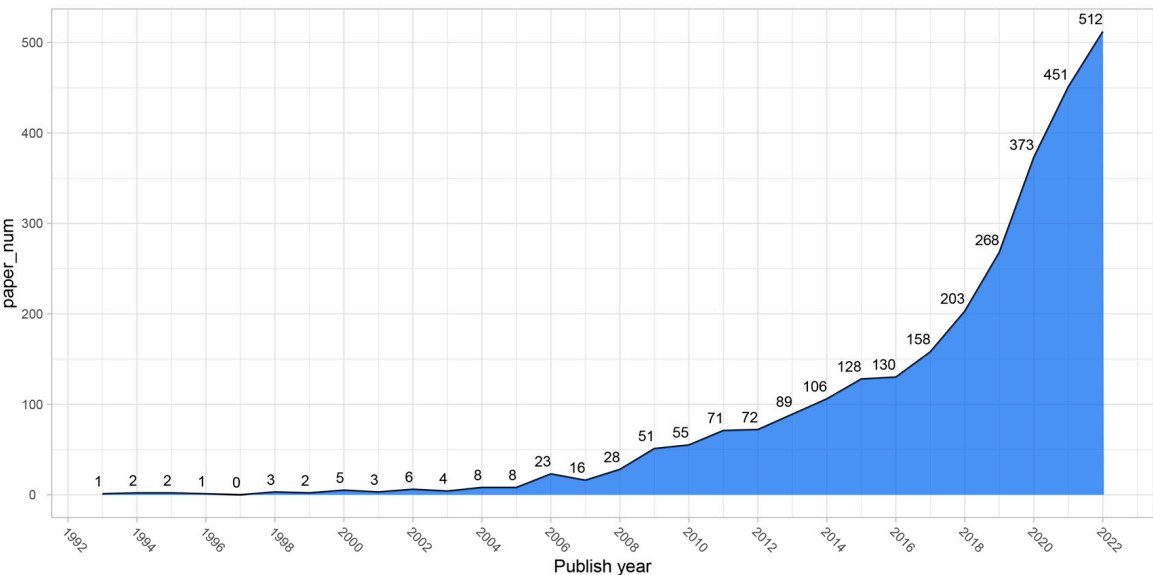

**Figure 2.** The growth trend of the related literature.

Based on the chronological chart of the growth of the literature on education for sustainable development, the research evolution can be divided into three phases:

**Emerging (1992–2004)**: A total of 37 papers were published in this phase, with an annual average of 2.85 studies. The first study appeared in 1993, and the number of relevant studies did not increase much until 2005. This period lasted for 13 years, and the number of publications per year was below 10, which was always at a low level. Since the introduction of the concept of sustainable development, the whole academic community has been in the receptive stage, and awareness and application in the field of education has lagged even more; this result is basically consistent with previous related studies.

**Development (2005–2015)**: A total of 647 literature articles were published during this phase, with an annual average of 58.82 studies, peaking in 2015 (128 articles). Although the reasons for the growth of the literature are manifold, one of the main reasons is the convergence of interest in ESD across the globe. This period marks the UN's Decade of ESD, with the majority of articles on research related to the integration of sustainable development principles and practices into all aspects of education. These themes reflect the goals of the decade, which were to encourage changes in knowledge structures, values, and attitudes, and to work toward a more sustainable and equitable society for all. Beginning with this period and continuing into the following years, ESD has generated widespread interest among researchers and the public media. A series of research and awareness campaigns were initiated on ESD due to the need for sustainable development.

**Maturation (2016–present)**: There were a total of 2095 publications in this phase, with an annual average of 299.29 publications. A simple explanation for this rapid growth may be the United Nations Summit held in New York in 2015, where all member states drafted and adopted the 2030 Agenda for Sustainable Development. As a result, there was an explosive growth of related papers after 2015, which generated global attention. Research in this phase focused more on the role of education in achieving sustainable development, especially from the perspective of Sustainable Development Goal 4 (SGD4) of quality education for all.

This growth trend is only a quantitative characteristic of researchers' contributions to research information, and it is not complete. The scientific publications in this field are diverse in type and coverage, and an analysis of the types of literature and their coverage is necessary to understand the contribution of each field to its development and the great efforts of researchers in each scientific field and discipline. We searched for five important types of literature, the most common being journal articles (2236 articles), which comprised 80.46% of all publications. The other literature types in order of proportion were proceedings papers (11.41%), editorial materials (6.19%), and reviews (1.94%). The top 10 research areas covered by ESD-related literature were: education and educational research (1592 articles); management (919 articles); sustainability science (641 articles); environment science (508 articles); economics (255 articles); political science (116 articles); and business, human geography, psychiatry and psychology, and sociology (all under 100 articles). In comparison with the period from 1992 to 2002, ESD research was mainly focused on Environmental Science, Political Science, and Human Geography, with a small proportion of education-related topics. The vast majority of articles are from the social sciences, while a single paper is documented in multiple fields of research, and interdisciplinary research is prevalent in this field of study. In addition, interdisciplinary fields were increasingly relevant to the research of ESD. Issues related to sustainability are complicated, and a single research field cannot figure out all of the problems. In fact, all scientific fields contributed to ESD research, thereby opening up new avenues for integrated research.

As for the research methods of the literature, given the large volume of data, this study used random sampling to code, classify, and analyze the sample. To achieve a 95% confidence interval, 556 pieces of literature were selected based on the relevant literature requirements [40], and three researchers were invited to select 200 pieces each randomly for statistical analysis; the percentage of data in the four groups did not fluctuate significantly. According to the results, 37.3% of the total literature was theoretical research, 56.4% was empirical research, and 6.3% comprised reviews. The distribution was relatively reasonable. Meanwhile, the analysis of empirical research methods revealed that the ESD-related studies were mainly qualitative (47.3%), while quantitative and mixed studies accounted for 34.6% and 18.1%, respectively. The proportion of quantitative studies was lower than expected, which may be related to the dynamic changes in the definition of sustainable development; furthermore, researchers generally pay close attention to the development of theories and concepts in this field. Comparing the relevant studies in the period from 1992 to 2002, it was observed that the studies mainly focused on policy evaluation and implementation, ESD theoretical constructs, etc. Empirical studies and quantitative studies were scarce, which also supports the previous hypothesis. Although there is no definition of a good or bad method, a mature body of knowledge needs to incorporate more analytical methods to cope with the changing social environment and to improve the effectiveness of the methods. The use of qualitative and simple quantitative methods in empirical research is a limitation of the field at this stage of development.

*4.2. Research Forces*

4.2.1. Country/Region Distribution

When collecting the number of national (regional) publications, England, Scotland, Northern Ireland, and Wales were included in the UK, the Federal Republic of Germany was included in Germany, and Hong Kong, Macao, and Taiwan were not included in China.

Based on this standard, publications from the literature came from 143 countries/regions, indicating a wide range of interests and widely distributed scientific results. The top 10 countries/regions of publication are shown in Figure 3.

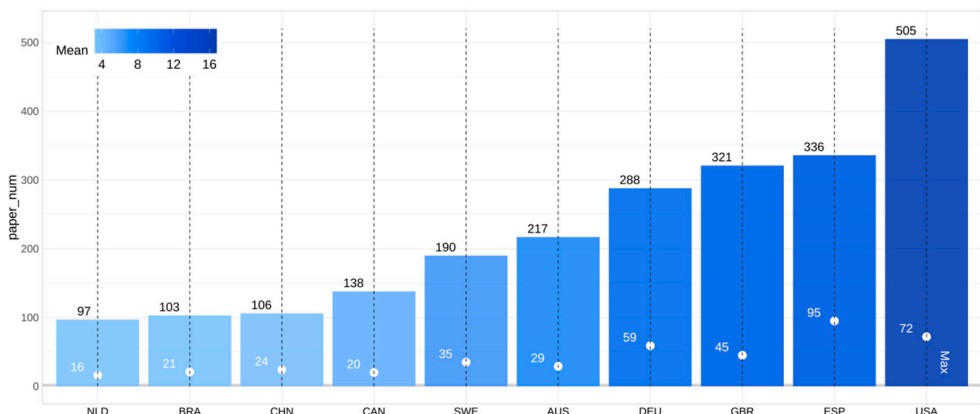

**Figure 3.** The distribution of ESD-related publications.

It is essential to note that no country has achieved all of the SDGs since 2015. Among the top 10 countries in terms of the number of articles published, Sweden is the closest to achieving the SDGs with a Sustainability Index score of 85.2 (the third in the world), while the rest of the countries are located behind the 10th [41]. There is no doubt that studies from the United States (505 articles), Spain (336 articles), and the United Kingdom (321 articles) have the greatest impact on the scale of research, with an average annual number of articles greater than 10 and a total number of articles accounting for 41.81% of the total number of articles. Meanwhile, the above countries are also ahead of other countries in terms of the growth rate of ESD-related articles.

### 4.2.2. Institutional Distribution

The total number of research institutions collected in this research was 2372, the vast majority of which are universities or research institutions. The top 10 institutions in terms of number of articles were: Arizona State University (61 articles), the Royal Melbourne Institute of Technology (57), the Leuphana University of Lüneburg (42 articles), Manchester Metropolitan University (39), Queensland University of Technology (38), the Universitat de València (36), James Cook University (33), Michigan State University (32), the University of Plymouth (31), and the University of Tasmania (31). The top 10 institutions contributed 14.4% of the articles, which confirms their importance for ESD research.

### 4.2.3. Authors Distribution

According to the database, there were 7253 related authors in total, and 27 authors with 10 or more publications. There was a large number of authors, but not many of them have published abundant research articles. On the one hand, this indicates that ESD research is widely followed worldwide, and on the other hand, it also indicates that ESD research has not formed an "academic monopoly," and that researchers from all over the world are contributing to ESD research. The top 10 authors were: Walter Leal Filho (30 articles, accounting for 1.08%), Niklas Gericke (25 articles, accounting for 0.90%), Matthias Barth (22 articles, accounting for 0.79%), Rodrigo Lozano (17 articles, accounting for 0.61%), Don Huisingh (15 articles, accounting for 0.93%), Helen Kopnina (14 articles, accounting for 0.50%), Ian Thomas (14 articles, accounting for 0.50%), Marco Rieckmann (14 articles, accounting for 0.50%), Jordi Segalas (14 articles, accounting for 0.50%), and Gisela Cebrián (14 articles, accounting for 0.50%). This ranking shows the active authors in the field of ESD; these authors are the scholars who have a long-term interest in ESD, and they are also the key figures making research strategies in this field.

In addition to the number of authors, the distribution of authors should examine the academic groups. Analyzing author collaboration networks provides insight into major academic teams and their research interests, and this enhances researchers' awareness of the positive role of academic groups in relation to research continuity and innovation. By using R software to visualize the literature's collaboration information, a knowledge map of author collaboration networks for sustainability education research was obtained, including 61 network nodes. Figure 4 shows the author groups with high collaboration frequency. It is formed from an academic group centered on four highly productive authors, including Walter Leal, Don Huisingh, Matthias Barth, and Niklas Gericke. They mainly conducted research on higher education [42], environmental education [41], future education [43], educational technology [44], and educational leadership and management [45]. Based on the overall performance and research areas of the collaborative groups, it was basically determined that these four collaborative groups are the current mainstream research teams in ESD.

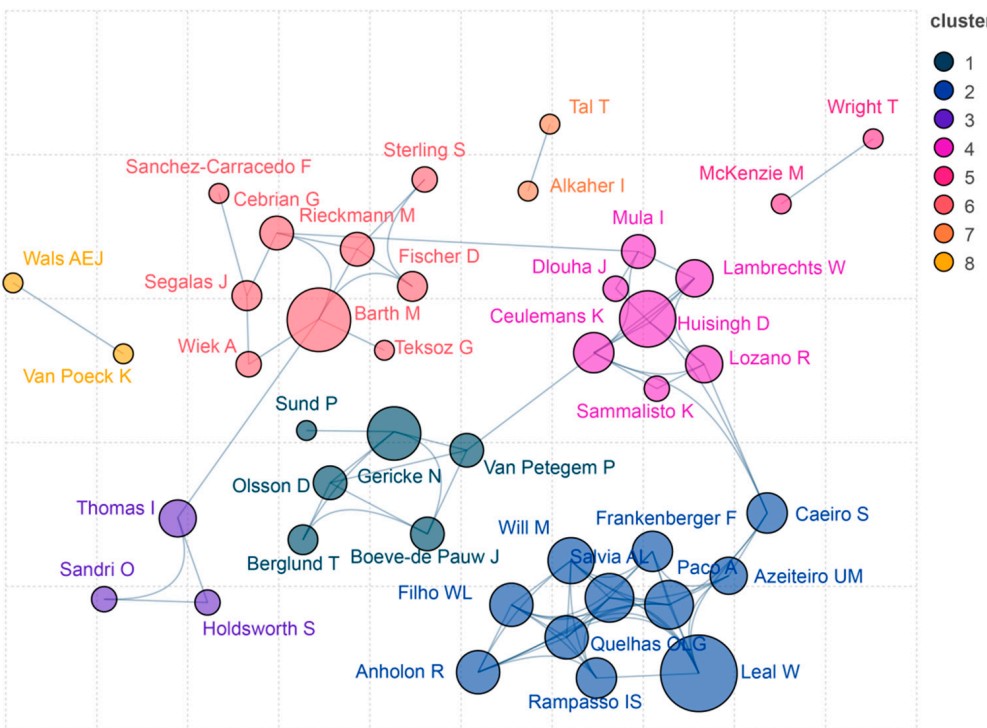

**Figure 4.** Author collaboration network map.

4.2.4. Journal Distribution

The analysis of journal distribution is essential for interdisciplinary field research to identify influence areas of the research. In total, 2779 journal articles were distributed across 491 different journals. On the positive side, the broad dispersion pattern suggests that research is not limited to a few professional journals due to the multidisciplinary attention. The journals covered topics including education, higher education, educational policy, educational administration, geography, science, the environment, engineering, energy, chemistry, and architecture (not fully listed), and there were many other specialized journals. In this study, the top 10 journals in terms of the total number of articles published, the impact factor, the classification of research, and the interdisciplinary dissemination characteristics of the research are highlighted (see Table 1).

**Table 1.** The top 10 professional journals related to ESD.

| Journal | IF | Subject and Quartile | Count | Citations |
|---|---|---|---|---|
| *SUSTAINABILITY* | 3.889 | ENVIRONMENTAL SCIENCES, Q2 ENVIRONMENTAL STUDIES, Q2 GREEN AND SUSTAINABLE SCIENCE AND TECHNOLOGY, Q3 | 745 | 3274 |
| *INTERNATIONAL JOURNAL OF SUSTAINABILITY IN HIGHER EDUCATION* | 4.120 | EDUCATION AND EDUCATIONAL RESEARCH, Q1 GREEN AND SUSTAINABLE SCIENCE AND TECHNOLOGY, Q3 | 358 | 5060 |
| *ENVIRONMENTAL EDUCATION RESEARCH* | 3.725 | EDUCATION AND EDUCATIONAL RESEARCH, Q1 ENVIRONMENTAL STUDIES, Q2 | 249 | 4563 |
| *JOURNAL OF CLEANER PRODUCTION* | 11.072 | ENGINEERING, ENVIRONMENTAL, Q1 ENVIRONMENTAL SCIENCES, Q1 GREEN AND SUSTAINABLE SCIENCE AND TECHNOLOGY, Q1 | 226 | 6566 |
| *JOURNAL OF ENVIRONMENTAL EDUCATION* | 2.975 | EDUCATION AND EDUCATIONAL RESEARCH, Q2 ENVIRONMENTAL STUDIES, Q3 | 44 | 1083 |
| *SUSTAINABILITY SCIENCE* | 7.196 | ENVIRONMENTAL SCIENCES, Q1 GREEN AND SUSTAINABLE SCIENCE AND TECHNOLOGY, Q2 | 34 | 1098 |
| *INTERNATIONAL JOURNAL OF MANAGEMENT EDUCATION* | 4.564 | EDUCATION AND EDUCATIONAL RESEARCH, Q1 MANAGEMENT, Q3 | 33 | 349 |
| *JOURNAL OF GEOGRAPHY IN HIGHER EDUCATION* | 1.727 | EDUCATION AND EDUCATIONAL RESEARCH, Q4 GEOGRAPHY, Q3 | 32 | 293 |
| *INTERNATIONAL JOURNAL OF ENGINEERING EDUCATION* | 0.971 | EDUCATION, SCIENTIFIC DISCIPLINES, Q4 ENGINEERING, MULTIDISCIPLINARY, Q4 | 27 | 180 |
| *RESEARCH IN SCIENCE EDUCATION* | 2.469 | EDUCATION AND EDUCATIONAL RESEARCH, Q2 | 26 | 524 |

*4.3. Research Themes*

Traditional reviews usually use various methods, such as grounded theory and content analysis, to analyze research themes, which reduce complex theoretical structures to indicators in an overly general way, producing decontextualized results and requiring a lot of professional background and interdisciplinary knowledge. They also consume a lot of time and energy and make it difficult to conduct large-scale data analysis. The LDA (Latent Dirichlet Allocation) theme model reduces subjective bias in large text corpus analysis, helps to discover potential themes instead of forcing them into the researchers' framework, and improves the way of thinking and explanation of the researchers. Manual coding was selected to abstract thematic keywords into higher-order thematic concepts to iterate existing theories with LDA themes. In this way, the validity of LDA thematic models has been improved, and the construction of new theories has been facilitated as well. Therefore, a mixed approach of LDA theme modeling and manual coding was used for theme identification of ESD. The LDA model is the most commonly used and matured theme analysis model; it is a Bayesian probabilistic model built from three levels of parameters: words, themes, and documents. In general, the process of producing LDA can be represented by the joint distribution of random variables, as shown in Equation (1).

$$p(\omega, z, \theta_m, \varphi_k | \alpha, \beta) = \prod_{n=1}^{N} p(\theta_m | \alpha) p(z_{m,n} | \theta_m) p(\varphi_k | \beta) p(\omega_{m,n} | \theta_{z_{m,n}}) \tag{1}$$

In this equation, $Z$ refers to the document topic distribution, $\theta$ refers to the proportion of the corresponding topic, $\Phi$ represents the topics of the article, and $\alpha$ and $\beta$ are two hyperparameters, which are generally regarded as default values [46]. $K$ refers to the optimal number, and it is determined by considering the Coherence Score and the Perplexity Score. The Coherence Score is used to calculate the semantic similarity, and a higher Coherence Score indicates a more interpretable model. The Perplexity Score is an index of cross-entropy, and a smaller value indicates a better model fit. According to the distribution of the Coherence Score and the Perplexity Score, 12 research themes were identified (the Perplexity Score showed a large decline when the number of themes was 12, and then it continued to decline, but it did not have a big change, and 12 themes were finally selected by combining the Perplexity Score with the distribution of the Coherence Score (shown in Figure 5). The two authors conducted secondary coding for the 12 themes based on the distance between the LDA theme maps and the understanding of keyword meanings and ESD. Finally, five research themes were identified (shown in Figure 6). The secondary coding was finished jointly by two PhD students in the field of education, and a new member was introduced to discuss and analyze the coding results for which there was disagreement until a consensus was finally reached. Finally, five research topics were identified based on the distance of the visualization mapping and the keyword lexical meaning.

**Theme I: Foundations of ESD Research**. This theme was an important cluster that ran throughout the research cycle and addressed numerous research topics, including definitions, concepts, theories, roles of sustainable development, and the global agenda for ESD and policies for integrating education into sustainable development. The concept of sustainable development has evolved from a simple methodological approach to an environmental–ecological construct, and, further, to a balance between the environment and development; it has finally reached a new concept of human rights as a priority [47]. ESD encompasses a novel vision of education that aims to empower people [48]; meanwhile, it provides education and training in citizenship and allows students to become agents of change, making it possible for everyone to acquire the knowledge, skills, attitudes, and values needed to shape a sustainable future. Therefore, there is an urgent need to examine how to better understand the interplay between ESD and the 2030 Agenda for Sustainable Development framework in a particular context [49]. In addition, researchers have also explored the relevant theoretical foundations of ESD, including educational ecology and educational anthropology from the perspective of ecological civilization [50,51], and related capital theories from the perspective of resource sustainability, such as natural capital [52] and social capital [53]. Moreover, the rapid emergence of ESD relies on the impetus of policy. National policies have been oriented to making targeted recommendations based on the social, economic, cultural, and environmental contexts of the region. The proposal of sustainable development policies is not only based on the current situation, but it also focuses on the implementation effect and later optimization, which is the solution to current problems and, moreover, the basis of future development.

**Theme II: Environmental Education**. This theme encompasses key perspectives on environmental education, including explaining the important role of ecological pedagogy in citizenship education, empirical insights on integrating key environmental issues, such as global climate change, biodiversity loss, pollution, waste disposal, etc., into the curriculum, and the construction of ecological spaces for education. Sustainability issues are deeply embedded in the inherent complexity of social–ecological systems, and environmental education involves a multitude of stakeholders who collaborate in research implementation spaces where science, policy making, local culture, and the environment intersect, and where environmental education often struggles in these productive but complex spaces [54]. Environmental education is an approach to ESD that has received particular attention in the field of educational ecology [55]. Environmental education is more like a conservation strategy that creates synergistic spaces to gather scientists, policymakers, community members, and other stakeholders to encourage intergroup research interactions [56]. En-

vironmental education is no longer simply a linear path from environment to education; instead, it emphasizes the dynamics, complexity, systemic nature, and sustainability of educational behavior.

**Theme III: Higher Education for Sustainable Development**. Higher education is considered the key to addressing the SDGs, and it has become a strategic actor in the development of sustainability [57]. Researchers have taken the SDGs as the context for analysis, proposing universities as the motivators of development and analyzing the challenges and innovative breakthroughs universities face in achieving sustainable development [58]. Getting learners to acquire sustainable competencies is considered a necessary activity for universities, and some studies have also analyzed the integration of sustainable development into higher education curricula, calling on teachers to adopt an interdisciplinary perspective to raise students' awareness of ESD and to raise awareness of sustainable development [59]. As an education and training center for educators, universities need to change their educational roles, organizational structures, and leadership models in response to new requirements to develop teachers' concepts of sustainable development and to achieve teachers' sustainable development [60]. In addition, universities need a process of review and renewal so that they can become direct leaders of change and implement their commitment to sustainability [61].

**Theme IV: Sustainable Development Capacity**. ESD is not only about behavioural change, but also about developing the knowledge and capacity needed to address sustainable development issues, which are the core of sustainable development [62]. Individuals should have the ability to consider sustainability issues and their possible solutions at different levels, such as the environmental level, economic level, social level, etc. This includes a comprehensive knowledge base of the origins, causes, and impacts of current problems and conflicting interests among stakeholders [63]. ESD breaks away from the traditional definition of education and is less about the transfer of specific knowledge but more about the ability to acquire and apply knowledge in a holistic manner. ESD activities cover all types of education, including formal, non-formal, general, and vocational education. ESD also covers all disciplines and all ages of education [64]; meanwhile, it goes beyond the transfer of knowledge and awareness raising to focus on building capacity for sustainable development. Capacity here refers to systems thinking capacity, performing capacity, strategic capacity, interpersonal capacity, and innovative learning skills [65–67]. In addition, sustainable development capacity has been integrated into teacher education, particularly in exploring teachers' perceptions of sustainability competencies, evaluating training programs and curricula, and assessing teachers' sustainable development capacity [68].

**Theme V: Educational Technology Innovation**. With the increasing instability, complexity, and uncertainty of the environment, educational technology innovation is undoubtedly a direction and requirement for the future of sustainable education. Nowadays, innovative experiences in the field of information and communication are integrated into ESD curricula to target sustainability issues [69], and how sustainability awareness can be enhanced through educational model design in an era of digital strategies is examined [70]. Meanwhile, the critical role that digital education can play in the future for sustainability capacity, environmental education, and ecosystem building and integration is explored [71]. Due to the permeability of digital technology, inclusive human development has been promoted [72], and the integration of technology and education becomes an effective means to promote sustainable development [73]. The future of education is a competition of resources and technologies, and a green and sustainable environment has a strong positive correlation with educational competitiveness. In other words, educational technology is a key element of education for sustainable development [74].

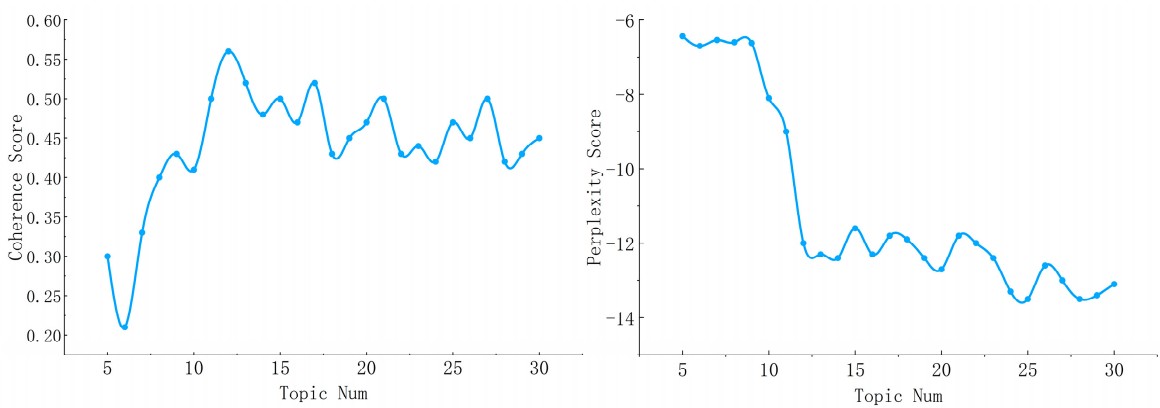

**Figure 5.** Coherence Score and Perplexity Score.

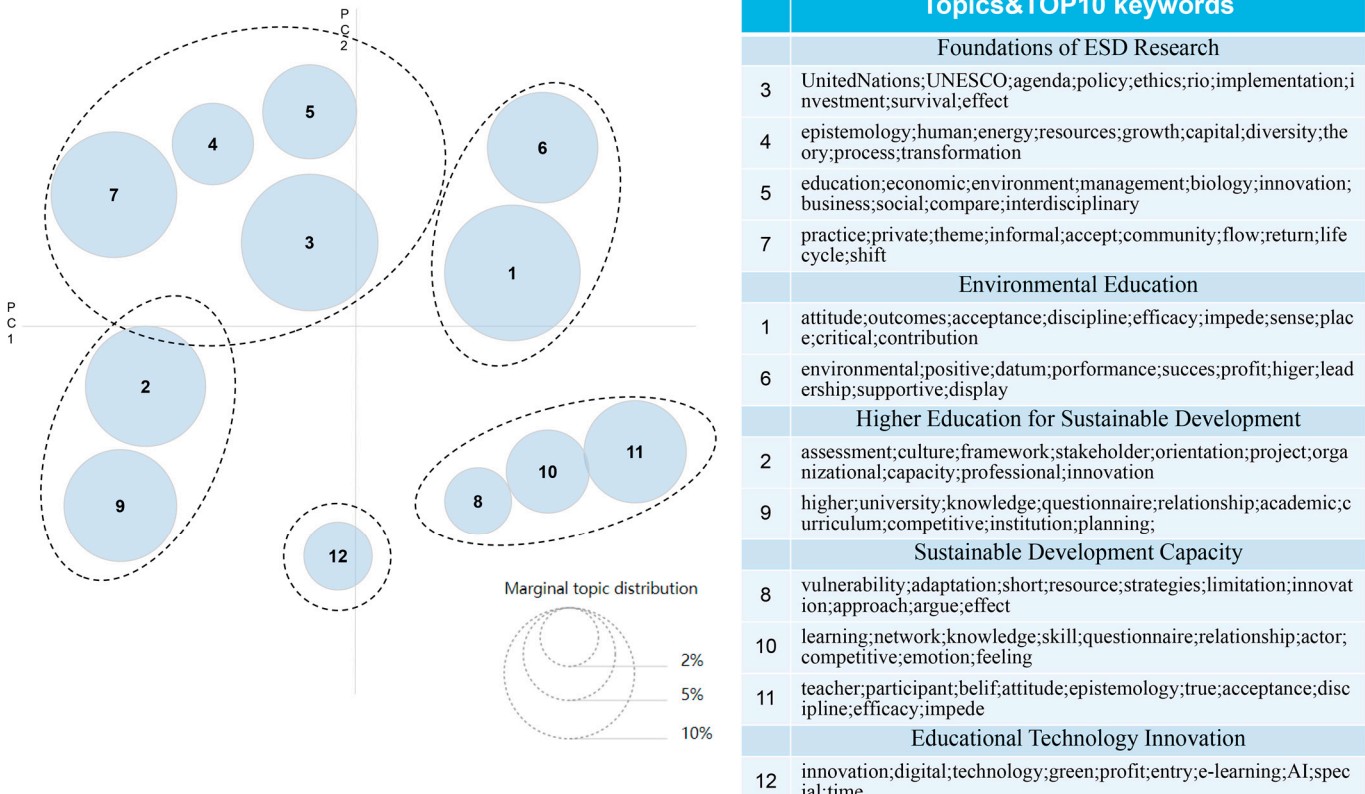

| | Topics&TOP10 keywords |
|---|---|
| | **Foundations of ESD Research** |
| 3 | UnitedNations;UNESCO;agenda;policy;ethics;rio;implementation;investment;survival;effect |
| 4 | epistemology;human;energy;resources;growth;capital;diversity;theory;process;transformation |
| 5 | education;economic;environment;management;biology;innovation;business;social;compare;interdisciplinary |
| 7 | practice;private;theme;informal;accept;community;flow;return;life cycle;shift |
| | **Environmental Education** |
| 1 | attitude;outcomes;acceptance;discipline;efficacy;impede;sense;place;critical;contribution |
| 6 | environmental;positive;datum;porformance;succes;profit;higer;leadership;supportive;display |
| | **Higher Education for Sustainable Development** |
| 2 | assessment;culture;framework;stakeholder;orientation;project;organizational;capacity;professional;innovation |
| 9 | higher;university;knowledge;questionnaire;relationship;academic;curriculum;competitive;institution;planning; |
| | **Sustainable Development Capacity** |
| 8 | vulnerability;adaptation;short;resource;strategies;limitation;innovation;approach;argue;effect |
| 10 | learning;network;knowledge;skill;questionnaire;relationship;actor;competitive;emotion;feeling |
| 11 | teacher;participant;belif;attitude;epistemology;true;acceptance;discipline;efficacy;impede |
| | **Educational Technology Innovation** |
| 12 | innovation;digital;technology;green;profit;entry;e-learning;AI;special;time |

**Figure 6.** LDA theme modeling and research keywords.

## 5. Discussion and Conclusions

In order to obtain the full picture of the research profile and main research themes of ESD in the past 30 years since the promulgation of UN Agenda 21, 2779 documents from the literature were collected and analyzed. Based on the results of this study, it is clear that ESD-related research has grown continuously over the past 30 years, and it is divided into three stages. These stages reflect the awareness development process of ESD and the efforts to promote sustainable development. Moreover, the introduction of the 2030 Agenda for Sustainable Development in 2015 has brought an explosive growth in ESD research. In addition, this study also conducted statistical analyses of the disciplinary distribution and methodological diversity of ESD and found that ESD research has obvious interdisciplinary characteristics. These characteristics bring a broader and more diverse research perspective to ESD research. However, the specific research methods used in ESD

research have obvious shortcomings; that is, the research methods used at this stage are mainly qualitative, while quantitative and mixed research methods are lacking. The need to incorporate more complex research designs [75] and to select higher-level research models and tools [76] remains a challenge for future ESD research.

This study also analyzed the main ESD research forces, including national distribution, institutional distribution, the main authors, and the main journals. The results show that ESD research is receiving attention from all countries around the world, but it is mainly concentrated in English-speaking countries, such as the United States, the United Kingdom, and some other developed European countries. This finding validates most of the related research [40,77,78]. At the same time, the results also indicate that less attention has been paid to ESD in developing regions, such as the Middle East and Africa. Although sustainable development issues and policies are primarily proposed and implemented by the Global North, sustainable development is a global issue that requires the involvement of the Global South, and countries belonging to the Global South need to understand and identify with ESD more than ever. In other words, increasing research interest in ESD outside of traditional academic research centers remains an urgent priority. Nowadays, the amount of research contributed by certain countries, such as China, Malaysia, Mexico, and Brazil, is growing. However, more research interest in EDS should be stimulated, and developing countries should be encouraged to make a more positive response. The uneven and concentrated distribution of publications is related to the different levels of development and the cultural backgrounds of each country, and there are great differences in the understandings of education for sustainable development among countries. Although sustainable development is a global concern, OECD member countries have shown more research attention in this area than, for example, countries in the Asia-Pacific region or sub-Saharan Africa. This research is concentrated in a small group of economically developed Western societies, and, more specifically, only 15.7% of the literature is led by developing countries. This may be due to the fact that developing countries have received less state support to conduct this kind of research, while research in OECD countries has involved more universities and research institutions and has continued to introduce relevant policies in the area of sustainable development. The cultural, social, and economic gaps between policy sources and implementation goals result in different research concerns and research challenges, and ESD requires a broader cultural, institutional, and socioeconomic context. Increasing the density of ESD beyond traditional academic research centers remains an urgent priority. Meanwhile, developing countries face resource scarcity that leads to a survivalist mentality and prompts policymakers to focus less on the future. ESD raises citizen awareness of sustainable development and mitigates the effects of debt through innovative curricula, progressive pedagogy, sustainability-oriented teaching materials, and monitoring and auditing mechanisms for teaching sustainable development. Fortunately, ESD research by scholars from developing countries (e.g., China, Brazil, Mexico, and South Africa) has also shown growth in the last three decades. Thus, there is a need to further stimulate interest and to develop capacity to support researchers in developing countries to engage in research, to respond positively, and to contribute to global efforts.

In terms of research institutions, most of them were universities, which confirms that higher education is the main site for the implementation of ESD concepts. Each university has its own research focus on sustainable development, which is reflected in the curriculum and training objectives. The institution with the most publications was Arizona State University, the first teacher training university in Arizona, which has been advocating for education for the future, environmental education, experiential education, and training future scholars in the field of sustainability by launching a graduate-level course on research design and methods for sustainable projects [79,80]. In addition, Arizona State University has recently launched a research initiative on "garden education" and is leading the field, surpassing the Royal Melbourne Institute of Technology University (RMIT) in the number of publications in 2017. "Garden Education" not only provides children and youth with the opportunity to explore the natural world, but it also gives

young people the opportunity to develop a wide range of academic and social skills and prepares them for future educational leadership for social change by developing specific sustainability competencies [81]. Australian institutions are prominent in the rankings, with four institutions dominating the top 10. Australia's agenda setting and specific actions around sustainability and sustainability education have made sustainability education one of three cross-curricular priorities for all areas of learning and at all levels of the Australian education system [82]. Much research is required to effectively embed ESD. RMIT is committed to embedding sustainability principles and practices into learning, teaching, research, and operational activities, and it has a number of highly productive authors [83,84]. Queensland University of Technology, on the other hand, is deeply engaged in research on sustainability education for children, exploring early childhood teacher education policy and practice in Australia [85]. The upward trend in Australian research since 2000 is a clear indication of the commitment of Australian universities to integrate sustainability into a variety of programs (e.g., the Master of Global Development Program at James Cook University) [86]. The articles from Australian authors covered a variety of topics important to the Australian curriculum: teachers' perspectives on sustainability, curriculum and teaching practices, mixed learning strategies for ESD, institutional changes for ESD at the university, ESD policy development and implementation, etc. Sustainability initiatives should occupy a strategic position in higher education institutions, where the mission of universities is to promote sustainable development and to attract researchers from institutions around the world for joint research through teaching, research, and based on publications [87]. The case of Australia also shows that universities do serve as important role models for other institutions in the implementation of sustainability programs in the academic field.

As for the distribution of the main authors, according to the data analysis, 10 core authors in the ESD research field were identified, and they kept focusing on their own research areas, forming four main research groups. The construction of an academic community has contributed significantly to the research, but the drawbacks are also obvious. Each research community has its own area of concentration, and there is a lack of cooperative communication among groups. Although there is active scholarly activity within the cooperative groups, inter-team cooperation may be more conducive to the establishment of new methodologies and theories. Walter Leal Filho is the most productive author. In the field of ESD, Walter is the founder of the European Academy of Sustainable Science and Research and the University Sustainability Program. Moreover, Walter founded the *International Journal of Sustainability in Higher Education*, which is the earliest journal focusing on sustainability under the higher education background. Each of the top 10 authors has continued to work in the field of sustainability education, conducting research on educational implementation programs, innovative teaching methods, and future educational governance in the context of sustainability. For example, Matthias emphasizes capacity development, innovative learning environments, and curriculum reform [44,83,88]. Niklas Gericke focuses on Swedish education practices for sustainable development and related resource allocation at the basic education level [45,89]. Ian Thomas focuses on the implementation of environmental education and education for sustainable development in Australian universities, as well as the development of graduate student capacity for sustainable development [90–92].

In this study, the core journals were identified in the ESD research field. These journals shape the ESD discourse and disseminate ESD ideas and concepts. Although the results overlap with some of the studies [93], this study demonstrates the interdisciplinary attributes of ESD research and the conservative distribution strategies from the perspective of journals' disciplinary affiliations and partitioning. These ten journals can be considered the key journals in the research field, publishing 63.84% of the articles; they are mainly situated in Q2 and Q3. Although there is some research published in Q1 journals, the overall publication strategy is relatively conservative. This publication strategy may facilitate the presentation of research results. Overall, the vast majority of studies in the field met high quality standards. In addition, we also found that there are some Open Access (OA)

journals among the top 10 journals of ESD research, including *Sustainability* (745) and the *International Journal of Sustainability in Higher Education* (358). Open access is considered to be the best way to improve the flow of knowledge, and the number of OA journals, the subjects covered, the geographical coverage, and the citation advantage all reflect, to a certain extent, the strength and effectiveness of their knowledge dissemination strategies [94,95]. The strength and effectiveness of their knowledge dissemination strategies have made great contributions to academic communication and scientific research. Submitting to OA journals in ESD research may be a better choice for improving the dissemination efficiency of ESD research.

Machine learning and natural language processing techniques have been used in this study, which create new directions for large-scale literature reviews. Through the combination of LDA thematic models and manual coding, this study tried to show the knowledge structure and research focus of ESD research. We finally divided the main research areas of ESD into the following five categories: foundation of ESD research, environmental education, higher education for sustainable development, sustainable development capacity, and innovation in education technology. The foundation of ESD is conducive to researchers' comprehensive understanding of ESD and promotes the alignment of ESD theoretical research with practice. HESD and sustainable development capacity represent important ESD-related practices, including important sites for the implementation of ESD concepts and the focus of ESD practice development activities. Environmental education examines ESD through an ecological lens, drawing attention to the dynamic synergy between people, external space, and space. Innovation in educational technology, on the other hand, provides new directions for the future development of ESD from the perspective of resources as well as green development. In summary, the identification of themes provides an entry point for future research, saving the time needed to identify key themes in ESD. This study also detailed the process of implementing the LDA topic model for making literature reviews, which may inspire scholars to try this approach in future review studies.

## 6. Limitations and Recommendations

As previously mentioned, there are no perfect bibliometric studies. This study is no exception, as the coverage of the literature is limited and may exclude some relevant studies. Future research may consider using more databases to conduct the research, such as Scopus, ERIC, PsyInfo, and PubMed, and including other types of publications in addition to journal articles. Despite these limitations, this study provides a retrospective look at ESD research in a reliable, transparent, and objective manner, reinforcing the understanding that education plays a critical role in achieving the UN Sustainable Development Goals. Together, these developments, characteristics, and trends form an emerging picture of an interdisciplinary academic field with the potential to have a profound impact on policy and practice in the years to come.

According to the findings of this study, it is necessary to move research centers from developed countries, such as Europe and the United States, to developing countries. In this process, it is essential to maximize the level of international understanding in regional development education. Meanwhile, there must be clarity regarding the local core content of the issue to connect these problems to global issues. It should be encouraged to establish and form a system of international, regional, and national networks with a wide range of partners, involving both universities and civil society. Furthermore, providing action guidelines for governments to guide them in adopting a holistic, interdisciplinary approach to ESD implementation and integrating ESD with their national education policies and systems is also necessary. This can help developing countries achieve sustainable development, expand global ESD initiatives, and normalize the multi-centered action model to promote localized ESD development.

Moreover, it is also important to focus on the role of higher education in promoting ESD. From the analysis of research institutions and research themes, it is clear that universi-

ties have a crucial role in sustainable development. As education and training centers for educators, universities need to pay attention to the implementation of their sustainability education activities. School education plays an important role in shaping the values and behaviors of students, and schools need to promote education that reinforces the basic literacy of sustainable development.

For researchers who aim to promote ESD research development, it is important to familiarize themselves with key figures in ESD research and their academic groups, to understand the latest research trends in ESD, to determine the research organizations' core missions and research directions, and to develop a feasible research plan. At the same time, it is important to build a stable academic community, to keep working continuously and deeply in the research field, to invest the researchers' personal intellectual capital in the team, and to reduce the time cost of knowledge acquisition. When working with a team with a high professional level, it is necessary to promote knowledge sharing among team members. Realizing the exchange and sharing of knowledge among different knowledge subjects lays a solid foundation for saving research resources and the value added of intellectual capital.

Finally, in the context of the widespread use of digital technology in education, the value of technological advances should be recognized, and new technologies should be actively utilized. New information technologies can transmit sustainable development information faster and over greater distances than traditional methods, making it easier to disseminate methods of living and knowledge that are appropriate to local languages and cultural contexts and to share information with people in other regions. People can access the essence of knowledge, exchange research results, and innovate sustainable development models by integrating information from different geographic regions and disciplines. UNESCO needs to give full play to its unique advantages in promoting education informatization and leading the achievement of sustainable development goals, focusing on the integration of educational technology and ESD.

**Author Contributions:** Conceptualization, C.Y. and Q.X.; methodology, C.Y.; software, C.Y.; validation, C.Y.; formal analysis, C.Y.; investigation, Q.X.; resources, Q.X.; data curation, Q.X.; writing—original draft preparation, Q.X.; writing—review and editing, Q.X.; visualization, C.Y.; supervision, C.Y.; project administration, C.Y. All authors have read and agreed to the published version of the manuscript.

**Funding:** This research received no external funding.

**Institutional Review Board Statement:** Not applicable.

**Informed Consent Statement:** Not applicable.

**Data Availability Statement:** Data will be made available upon request.

**Conflicts of Interest:** The authors declare no conflict of interest.

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
