# Peer review of "A Bibliometric Review of Education for Sustainable Development, 1992–2022"

_sustainability, doi:10.3390/su151410823_

Round 1

Reviewer 1 Report

- A brief introduction/explanation of the word" bibliometric" needs to be added with valid citation.

- Being an integral component of bibliometric, citation analysis has not been given due weightage. To improve the quality of the paper, focus on citation analysis will improve quality of the paper.

- Double check the language of the text for minor corrections.

-   Cite some recent studies related to bibliometric.   

- Double check the language of the text for minor corrections.

Author Response

Dear Reviewer,

Thank you pretty much for your kind and valuable comments. They are important for us to polish our article. We try to make feedback for each comment as follows.

  1. A brief introduction/explanation of the word" bibliometric" needs to be added with valid citation.

We have explained the word “Bibliometric” and further describe the advantages and disadvantages of this method. Please see on page 2.

  1. Being an integral component of bibliometric, citation analysis has not been given due weightage. To improve the quality of the paper, focus on citation analysis will improve quality of the paper.

The article mainly explores the research progress of ESD as a whole from a macro perspective, and the study of citations and highly cited papers will be explored in detail and in depth in subsequent studies.

  1. Double check the language of the text for minor corrections.

We have double checked the language of the article and made corrections.

  1. Cite some recent studies related to bibliometric.   

We have added the most recent studies related to bibliometric and highlighted in the reference list.

We would like to show our appreciation again and your kind suggestions are important for us.

Best regards,

Chuang Yang & Qi Xiu

Reviewer 2 Report

In general, it is an interesting paper that provides an overview of the progress of ESD research based on the following Question 1: To examine the quantitative progress of ESD research, the interdisciplinary situation, and the main research methods chosen from a broader historical perspective.  

Question 2: Identify the distribution and characteristics of major research forces, including countries (regions), institutions and productivity of authors and journals, and further explore the reasons for their generation.  Question 3: Analyze the intellectual structure and identify key research areas.

Looking into the way the results are presented, the first question is not adequately answered. In any case, it will be more informative if the results are structured on the titles of each question. The assumption that for the main authors’ distribution, the collaboration network revealed that about 10 scholars largely from Europe contributed to the ESD research field, and they kept focusing on their own research areas, forming four main research groups is misleading. There are many other scholars with even more articles published who have at least equally contributed to the field. In addition to that, in the discussion, there is a need to explore more the implications of the study,

Author Response

Dear Reviewer,

Thank you pretty much for your kind and valuable comments. They are important for us to polish our article. We try to make feedback for each comment as follows.

Looking into the way the results are presented, the first question is not adequately answered. In any case, it will be more informative if the results are structured on the titles of each question. The assumption that for the main authors’ distribution, the collaboration network revealed that about 10 scholars largely from Europe contributed to the ESD research field, and they kept focusing on their own research areas, forming four main research groups is misleading. There are many other scholars with even more articles published who have at least equally contributed to the field. In addition to that, in the discussion, there is a need to explore more the implications of the study.

Cause research methods and interdisciplinary situation usually have smaller change over time, the article didn’t discuss them with too much details, but the article adds a comparison of the overall research methods and interdisciplinary situation with the initial decade on page 8. The discussion of the multidisciplinary character of the research methods and research has not changed much over time and the length of the discussion is not as detailed as the time, so the corresponding subheadings are not listed. From the overall analysis of the author network, more explanations have been added on page 10. The article acknowledges the breadth and contribution of the authors of ESD studies (7253 authors), and also points out the authors who have published more articles and the academic networks they have formed, suggesting the importance of the continuity building of research. Moreover, the article also points out the disadvantages of this development model on page 16.

We would like to show our appreciation again and your kind suggestions are important for us.

Best regards,

Chuang Yang & Qi Xiu

Reviewer 3 Report

Overall this was a very well written manuscript. I enjoyed reading the manuscript and feel as though this work contributes to the literature in a meaningful way. There were very few areas in the manuscript where I have significant concern; however, I have a few suggestions for improvement.

1 - It would be helpful to add some additional clarification to the methods section regrading the data analysis software tools. Specifically R and Python were used; however, it is unclear what packages were used for what analysis. 

2 - Additional description of the automated data collection and analysis may be helpful. It is a little unclear how the 4541 original documents were sorted down to 2779. Was this an automated process? If yes, was there any manual oversight, or review, associated with the removal of items? Either way, including more detailed inclusion, exclusion criteria may be helpful.

3 - If the majority of the analysis was completed using automated techniques, the packages and machine learning processes/training should be described in more detail. If the analysis only included sorting key words, titles, and authors this should be clearly stated. For example, even under these very basic analysis conditions how were author names matched? Is there a possibility for misclassification if author names are not identical from one publication to the next?

4 - A recommendation would be to include an additional limitation regarding the trade-off between automated analysis which can process large volumes of data at scale and manual analysis which allows for more nuanced analysis and synthesis of data across sources - allowing for themes to emerge more organically.  An additional recommendation would be to indicate the volume of publications should not necessarily serve as a proxy for impact. Looking at article citation data may be a stronger indicator of perceived impact in the discipline.

5 - Adding recommendations for future research and practice may help to elevate the work and make it more actionable. 

6 - Line 69, suggest updating the term 'Nowadays' term reads a bit colloquial

Author Response

Dear Reviewer,

Thank you pretty much for your kind and valuable comments. They are important for us to polish our article. We try to make feedback for each comment as follows.

1 - It would be helpful to add some additional clarification to the methods section regrading the data analysis software tools. Specifically R and Python were used; however, it is unclear what packages were used for what analysis.

We have added more explanation about the data analysis software tools and more details about the analysis method on page 5.

2 - Additional description of the automated data collection and analysis may be helpful. It is a little unclear how the 4541 original documents were sorted down to 2779. Was this an automated process? If yes, was there any manual oversight, or review, associated with the removal of items? Either way, including more detailed inclusion, exclusion criteria may be helpful.

The data were mainly obtained automatically by python, and the screening of the initial literature was mainly made by manual review, with specific screening criteria and strategies presented on page 5.

3 - If the majority of the analysis was completed using automated techniques, the packages and machine learning processes/training should be described in more detail. If the analysis only included sorting key words, titles, and authors this should be clearly stated. For example, even under these very basic analysis conditions how were author names matched? Is there a possibility for misclassification if author names are not identical from one publication to the next?

We added more explanation and details description on page 5. The statistical classification of the information of the literature such as keywords, countries, and authors was mainly analyzed by relying on the “bibiometrix” of R, which automatically filters, de-emphasizes, and unifies the naming of authors and publications, etc., with a low probability of misclassification.

4 - A recommendation would be to include an additional limitation regarding the trade-off between automated analysis which can process large volumes of data at scale and manual analysis which allows for more nuanced analysis and synthesis of data across sources - allowing for themes to emerge more organically.  An additional recommendation would be to indicate the volume of publications should not necessarily serve as a proxy for impact. Looking at article citation data may be a stronger indicator of perceived impact in the discipline.

The pros and cons of manual and automated analysis are explained on page 12. The number of publications is analyzed in order to illustrate ESD’s publication preferences and publication strategies, and citation information is presented for each journal. The citations of individual articles will be explored with specific detail in the coming subsequent research.

5 - Adding recommendations for future research and practice may help to elevate the work and make it more actionable.

We have provided some reference suggestions based on the findings of the study in terms of the transfer of research centers, the importance of higher education, the use of educational technology, and the formation of academic teams, please see page 17.

6 - Line 69, suggest updating the term 'Nowadays' term reads a bit colloquial.

We have changed “nowadays” to “recently”. Thanks.

We would like to show our appreciation again and your kind suggestions are important for us.

Best regards,

Chuang Yang & Qi Xiu

Reviewer 4 Report

1. Research questions or research problems should be added either on the introduction or literature review

2. The elaboration of the processes of screening, eligibility and included should be more detailed. I am concerned with the elimination processes. The questions over why and what requirements of the elimination must be informed from A to Z. This will confuse readers and can nor be theoritical reference for future studies.

3. Some statements like the use of R and Phython needs background checks, why did you choose both, the benefits and weaknesses should be elaborated. And how did you conduct the process matters for readers, please revise.

4. Practical and theoritical recommendations are missing in the end of the manuscript. You may add some elaboration in the limitation part and change the sub-heading into "limitation and recommendation".

5. I suggest for the data to be open in public DOI storage like Mendeley or Figshare. This is important to improve the citations of your work.

6. Hope you can revise the paper and have a great paper as an important academic reference for ESD. Congrats

Author Response

Dear Reviewer,

Thank you pretty much for your kind and valuable comments. They are important for us to polish our article. We try to make feedback for each comment as follows.

  1. Research questions or research problems should be added either on the introduction or literature review

In Introduction part, we explained the significance of the study in terms of the positive role of education for sustainable development and the lack of retrospective studies, please see on page 2. And in the literature review part, we raised specific research questions based on previous studies.

  1. The elaboration of the processes of screening, eligibility and included should be more detailed. I am concerned with the elimination processes. The questions over why and what requirements of the elimination must be informed from A to Z. This will confuse readers and can nor be theoritical reference for future studies.

On page 4, we have provided a search strategy form to add specific screening steps to the PRISMA system evaluation criteria for data screening (P5) (mainly talked about the type of initial literature, language, research range, and subject relevance).

  1. Some statements like the use of R and Phython needs background checks, why did you choose both, the benefits and weaknesses should be elaborated. And how did you conduct the process matters for readers, please revise.

We have added the background of why choosing the two software packages (open access, free, timely maintenance, etc.) on page 5. The specific process and the software packages selection have also been explained.

  1. Practical and theoritical recommendations are missing in the end of the manuscript. You may add some elaboration in the limitation part and change the sub-heading into "limitation and recommendation".

We have changed the sub-heading and add more content in this part.

  1. I suggest for the data to be open in public DOI storage like Mendeley or Figshare. This is important to improve the citations of your work.

The subsequent research work needs to rely on the original data, it is not a right time to disclose now. Appreciate for your kind understand.

  1. Hope you can revise the paper and have a great paper as an important academic reference for ESD. Congrats

Thank you so much for your kind encouragement, we will try our best to make it better.

We would like to show our appreciation again and your kind suggestions are important for us.

Best regards,

Chuang Yang & Qi Xiu
